# Machine Learning Model in Predicting Sarcopenia in Crohn’s Disease Based on Simple Clinical and Anthropometric Measures

**DOI:** 10.3390/ijerph20010656

**Published:** 2022-12-30

**Authors:** Yujen Tseng, Shaocong Mo, Yanwei Zeng, Wanwei Zheng, Huan Song, Bing Zhong, Feifei Luo, Lan Rong, Jie Liu, Zhongguang Luo

**Affiliations:** 1Department of Digestive Diseases, Huashan Hospital, Fudan University, Shanghai 200040, China; 2Department of Radiology, Huashan Hospital, Fudan University, Shanghai 200040, China; 3Department of Allergy and Immunology, Huashan Hospital, Fudan University, Shanghai 200040, China

**Keywords:** Crohn’s disease, sarcopenia, machine learning

## Abstract

Sarcopenia is associated with increased morbidity and mortality in Crohn’s disease. The present study is aimed at investigating the different diagnostic performance of different machine learning models in identifying sarcopenia in Crohn’s disease. Patients diagnosed with Crohn’s disease at our center provided clinical, anthropometric, and radiological data. The cross-sectional CT slice at L3 was used for segmentation and the calculation of body composition. The prevalence of sarcopenia was calculated, and the clinical parameters were compared. A total of 167 patients were included in the present study, of which 127 (76.0%) were male and 40 (24.0%) were female, with an average age of 36.1 ± 14.3 years old. Based on the previously defined cut-off value of sarcopenia, 118 (70.7%) patients had sarcopenia. Seven machine learning models were trained with the randomly allocated training cohort (80%) then evaluated on the validation cohort (20%). A comprehensive comparison showed that LightGBM was the most ideal diagnostic model, with an AUC of 0.933, AUCPR of 0.970, sensitivity of 72.7%, and specificity of 87.0%. The LightGBM model may facilitate a population management strategy with early identification of sarcopenia in Crohn’s disease, while providing guidance for nutritional support and an alternative surveillance modality for long-term patient follow-up.

## 1. Introduction

Sarcopenia is defined as the progressive generalized loss of skeletal muscle mass and strength associated with increased morbidity and mortality. Although more commonly recognized as a degenerative process in the older population, sarcopenia has also been associated with a wide range of disease spectrum, including cancer, metabolic disorders, and inflammatory diseases [1,2]. Malabsorption is a major contributor to muscle loss and dysfunction in sarcopenia and is commonly observed in patients with inflammatory bowel disease (IBD) [3].

Crohn’s disease (CD), a subtype of inflammatory bowel disease, is characterized by transmural inflammation of the gastrointestinal tract and can often be complicated by strictures, fistulae, and abscess formation [4]. Due to the different severity of enteropathy, CD patients have varying degrees of malnutrition resulting in loss of muscle mass and function. Previous studies have demonstrated that sarcopenia has a prognostic role in the management of inflammatory bowel disease. The estimated prevalence of sarcopenia in CD ranges from 20–70%, based on data collected in different patient subgroups, including active or stable disease, pediatric patients, and different ethnicity [5]. Sarcopenia is undoubtedly the direct result of chronic inflammation and malnutrition. Growing evidence has shown that it has a debilitating effect on patients’ physiological reserve, which hinders their ability for postoperative recovery and increases the likelihood of surgical complications. Alterations in body composition throughout the course of the disease may also provide therapeutic indications [6].

According to the European Working Group on Sarcopenia in Older People (EWGSOP) [1] and Asian Working Group for Sarcopenia (AWGS) [2], skeletal muscle mass can be quantified as appendicular skeletal muscle mass (ASM) or cross-sectional analysis of a specific muscle group based on magnetic resonance imaging (MRI) and computed tomography (CT), which is considered the gold standard for diagnosis. However, the requirement of specific image processing software to quantify CT images can be a labor-intensive and time-consuming process even for trained specialists.

Machine learning is a subdomain of artificial intelligence (AI), which incorporates the use of software algorithms to identify patterns in clinical datasets. Machine learning has rapidly driven the progress of AI in health care, demonstrating impressive results in patient monitoring, clinical decision support, improving diagnostics and prognostics, and even clinical research [7]. The incorporation of machine learning into clinical practice has become an area of interest for clinicians across all subspecialties.

The present study is determined to explore the incidence of sarcopenia in CD patients and to investigate the accuracy of different machine learning models and their performance in determining sarcopenia in Crohn’s disease based on easily accessible clinical data. Patients diagnosed with Crohn’s disease at a tertiary hospital in Shanghai, China, were recruited and evaluated for the presence of sarcopenia, which was determined based on skeletal muscle segmentation of the patient’s abdominal CT scan. Clinical and anthropometric data were collected and evaluated for their performance in identifying sarcopenia based on different machine learning models. Different machine learning models were tested based on randomly assigned training and validation sets. We aim to explore the clinical potential of an ideal diagnostic model that can be used not only to identify the presence of sarcopenia but also to provide a conventional surveillance modality throughout the course of the disease. The following machine learning methods, including Naive Bayes, Logistic Model, Classification Tree, Random Forest, adaBoost, XGBoost, and LightGBM, could successfully identify the presence of sarcopenia based on baseline characteristics. Among these models, LightGBM yielded the most ideal diagnostic performance (AUC of 0.933, AUCPR of 0.970, sensitivity of 72.7%, specificity of 87.0%), which portrayed its clinical applicability. Furthermore, we applied SHAP values to interpret the LightGBM model, highlighting the significance of BMI, gender, height, and CRP in the diagnosis of sarcopenia in Crohn’s disease.

## 2. Materials and Methods

### 2.1. Patients and Data Collection

The present study was conducted at the Department of Digestive Diseases of Huashan Hospital, Fudan University, a municipal tertiary medical center located in Shanghai, China. A retrospective review of our inpatient medical database was conducted. A total of 323 patients diagnosed and treated for Crohn’s disease from January 2016 through March 2022 were reviewed for eligibility. Patients were excluded if the abdominal or small intestinal CT was not performed at our center within one month of admission. Patients with missing or incomplete clinical data were also excluded. A total of 167 patients were ultimately included in the present study. Detailed patient history was reviewed and collected for demographic data, disease characteristics based on Montreal classification, laboratory examinations, endoscopic results with SES-CD (simple endoscopic score for Crohn’s disease), and body composition via radiological findings, prior to the diagnosis of sarcopenia. Candidate predictor values were identified, including gender; age; age at diagnosis (A1, A2, A3); disease location (L1, L2, L3, L4); disease behavior (B1, B2, B3); perianal disease; SES-CD; laboratory data including white blood cell counts (WBC), red blood cell counts (RBC), hemoglobin (Hb), platelet counts (PLT), albumin (ALB), prealbumin (PA), erythrocyte sedimentation rate (ESR), and C-reactive protein (CRP); height (cm); weight (kg); and body mass index (BMI, kg/m^2^).

### 2.2. Body Composition

Abdominal CT or small intestinal CT was routinely conducted as part of disease assessment for hospitalized patients. The original Dicom files were retrospectively reviewed by a blinded radiologist. A single cross-sectional CT slice at the level of L3 (third lumbar vertebrae) was used for segmentation of skeletal muscle mass (SMM), visceral adipose tissue (VAT), and subcutaneous adipose tissue (SAT). SliceOmatic software (5.0 Rev-9, Tomovision, Montreal, QC, Canada) was used to calculate body composition including skeletal muscle mass (SMM), visceral adipose tissue (VAT), and subcutaneous adipose tissue (SAT). The threshold for delineating skeletal muscle tissue was −29 to 150 Hounsfield units (HU), visceral adipose tissue was −150 to −50 HU, and subcutaneous adipose tissue was −190 to −30 HU. Manual segmentation was performed to calculate the surface area (cm^2^) of the region of interest, which included the psoas, quadratus lumborum, transverse abdominis, external and internal obliques, rectus abdominis, and erector spinae muscles. Skeletal muscle index (SMI), visceral adipose index (VAI), and subcutaneous adipose index (SAI) were subsequently calculated by dividing the targeted surface area by height squared. A cutoff value of SMI <49.9 cm^2^/m^2^ in male patients and <28.7 cm^2^/m^2^ in female patients was used to define sarcopenia, based on a previous report of a Chinese IBD population cohort [8]. 

### 2.3. Machine Learning and Statistical Analyses

Patients were randomly divided into training (*n* = 133) and validation cohorts (*n* = 34) in a ratio of 80% to 20%. Naive Bayes from e1071 package [9], Logistic Model [10], Classification Tree from rpart package [11], Random Forest from randomForest package [12], adaBoost from JOUSBoost package [13], XGBoost model from xgboost package [14], and LightGBM model from lightgbm package [15] were applied for machine learning. For XGBoost model, the optimal parameters including eta value, maximal depth, minimal child weight, and subsample were determined via 5-fold cross-validation and Bayesian optimization as performed in previous studies, and trained by 10 iterations [16]. For LightGBM model, the optimal parameters including lambda1, lambda2, feature fraction, and MinHessianLeaf were determined by grid search, and the training process was repeated by 100 rounds. For outcome output, Naive Bayes, Classification Tree, Random Forest, and adaBoost models directly output binary variables (sarcopenic or nonsarcopenic). Logistic Model, XGBoost, and LightGBM models output the probability that the patient had sarcopenia. A probability greater than 0.5 was predictive of sarcopenia. For the model evaluation, confusionMatrix was constructed by caret package. Furthermore, Matthews correlation coefficient (MCC) and F1 score were calculated for each model. The area under the receiver operating characteristic (ROC) curve and precision-recall (PR) curve was utilized to optimize model selection. To explain the variables in the XGBoost or LightGBM model, SHAP (Shapley additive explanations) value was calculated for each variate of each sample with shapforxgboost package, which was visualized by ggplot2. Continuous variables were compared by Student‘s T-test or Wilcoxon test, depending on the distribution and variance of the data. Categorical variables were tested using chi-square test or Fisher’s test. All statistical analyses were performed with R (4.0.3).

## 3. Results

### 3.1. Patient Characteristics

A total of 167 patients were included in the present study, of which 127 (76.0%) were male and 40 (24.0%) were female, with an average age of 36.1 ± 14.3 years old. Based on the previously defined cut-off value of sarcopenia, 118 (70.7%) patients had sarcopenia. Disease characteristics of Crohn’s disease were defined by the Montreal classification. The majority of patients (62.9%) experienced disease onset at the age of 17–40 years old (A2), followed by >40 years old (A3), and <17 years old (A1). Most patients had ileal or ileocolic disease (45.8% and 42.8%, respectively), while stricturing disease was the more common disease behavior (51.8%). Perianal disease was observed in the majority of patients (74.3%). All patients received endoscopic examination to assess disease severity via colonoscopy, retrograde double-balloon endoscopy, or both. SES-CD was used to determine the severity of mucosal defect. The average SES-CD score for all patients was 6.47 ± 5.9. Detailed patient characteristics are summarized in Table 1.

Baseline characteristics were compared between sarcopenic and nonsarcopenic patients. A significant difference was noted in gender distribution and age of disease onset. Sarcopenia had a male predominance (92.4%) and a younger age of disease diagnosis. A significant difference was also noted in WBC, RBC, and PLT. Anthropometric data showed that sarcopenic patients were taller, but their BMI was significantly lower than nonsarcopenic patients. Body composition data showed an average SMI of 39.4 ± 6.4 cm^2^/m^2^ in sarcopenic patients, while the average SMI was 41.6 ± 9.8 cm^2^/m^2^ in nonsarcopenic patients. Visceral adipose tissue and subcutaneous adipose tissue were also significantly lower in sarcopenic patients compared to nonsarcopenic patients, with an average VAI of 17.7 ± 14.7 cm^2^/m^2^ vs. 33.67 ± 19.9 cm^2^/m^2^ and SAI of 21.6 ± 13.8 cm^2^/m^2^ vs. 43.44 ± 19.0 cm^2^/m^2^, respectively (Appendix A).

### 3.2. Model Building and Evaluation

A total of seven machine learning models were implemented to determine an ideal diagnostic algorithm for sarcopenia, namely Naive Bayes, Logistic Model, Classification Tree, Random Forest, adaBoost, XGBoost, and LightGBM. All 18 variables (as listed in Section 2.1) derived from clinical and laboratory data were included for algorithm calculations. Each machine learning model was trained with the randomly allocated training cohort (80%) as described in Section 2.3. The performance of each model was evaluated on the validation cohort (20%), in which performance results including accuracy, sensitivity, specificity, precision, F1 score, MCC, AUC (area under curve), AUPRC (area under the precision-recall curve), TP (true positive), FP (false positive), FN (false negative), TN (true negative), PPV (positive predictive value), and NPV (negative predictive value) were calculated and summarized in Table 2. A comprehensive comparison showed that LightGBM was the most ideal diagnostic model, with an AUC of 0.933, AUCPR of 0.970, sensitivity of 72.7%, specificity of 87.0%, PPV of 0.727, NPV of 0.870, F1 of 0.727, and MCC of 0.597, while the adaBoost model was the least ideal (Figure 1).

Consequently, SHAP analysis was used to further interpret the results of the LightGBM model for its optimal diagnostic performance by computing the contribution of each variable. The average Shapley scores plot ranked the variables from most important to least important in contribution to the patient’s sarcopenic status in Crohn’s disease. The four most important variables were BMI, followed by gender, height, and CRP, based on the LightGBM model (Figure 2A).

The SHAP summary plot shows the distribution of all 18 variables and their corresponding positive or negative contribution to the prediction of sarcopenia. Each dot represents per patient per feature, colored according to an attribution value, wherein yellow represents a lower value, while blue represents a higher value. BMI has the greatest variability, in which a lower BMI value was associated with a positive Shapley value, contributing to an increased likelihood of sarcopenia. Conversely, a higher prealbumin level predicted a lower chance of sarcopenia due to a negative marginal impact on the Shapley value (Figure 2B).

The SHAP dependence plot represents the importance of each variable with respect to baseline. A corresponding Shapley value that exceeds one contributes to the risk of sarcopenia. In general, patients with lower BMI, taller stature, higher CRP, higher PLT, lower PA, lower WBC, younger age, and lower ALB contributed to an increased risk of sarcopenia (Appendix A).

The collective effect of each variable can be visualized at the local (patient) level Shapely values. Each bar represents the patient’s total Shapely score based on the additive contribution of each variable in a landscape view (Appendix A). 

Representative case studies show the performance of the diagnostic model at a local level. The Shapley values of the top four predictive variables indicate the risk of sarcopenia in individual patients, for instance, a taller male patient with a lower BMI and higher CRP is at risk for sarcopenia (red background) compared to a shorter female patient with higher BMI and lower CRP (blue background) (Figure 2C).

## 4. Discussion

The present study presents a diagnostic model for predicting sarcopenia in Crohn’s disease. The optimal algorithm was determined based on reruns of different machine learning models with easily accessible clinical and laboratory data. It was discovered that multiple machine learning methods, including Naive Bayes, Logistic Model, Classification Tree, Random Forest, adaBoost, XGBoost, and LightGBM could identify sarcopenia given baseline data. Among the models, LightGBM shared the best diagnostic performance on the validation cohort (ROC-AUC = 0.933, PR-AUC = 0.970, sensitivity = 0.727, specificity = 0.870), which indicated optimal clinical applicability. Furthermore, we applied SHAP values to interpret the LightGBM model, indicating the significance of BMI, gender, height, and CRP in the diagnosis of sarcopenia among CD patients. The high-performance gradient boosting framework displayed promising value in the clinical diagnosis of sarcopenia in Crohn’s disease patients. To the best of our knowledge, this is the first article to predict the presence of sarcopenia in Crohn’s disease patients based on a screening of different machine learning algorithms. The variables applied in the present study included demographic and anthropometric data, and laboratory parameters which are routinely collected upon hospital admission or at outpatient clinics. 

Machine learning has become a new tool used in the field of medicine, applied to medical diagnosis and clinical decisions. However, the application of machine learning from theory to practice still has a long way to go [17]. Traditional machine learning methods, such as logistic regression and random forests, are prone to underfitting due to the lack of boosting and ensembling. In this study, high-tech gradient boosting algorithms including XGBoost and LightGBM were integrated into the machine learning framework to establish a powerful and clinically applicable model [18]. Physicians may be reluctant to employ machine learning in clinical decision making due to the lack of transparency in the derivation of a diagnosis or decision. In order to improve the transparency of the calculation process, the Shapley additive explanations (SHAP) methodology provided a visual depiction of our predictions [19,20]. Highly relevant variables including BMI, height, gender, and CRP could easily identify patients with sarcopenia based on Shapley values at a local (patient) level.

Machine learning has weighted the role of different variables in determining the presence of sarcopenia. The identification of risk factors may provide insight into disease pathogenesis. The diagnosis of sarcopenia has been rapidly gaining awareness in various disease entities, including cancer, metabolic syndromes, and autoimmune disease. As a subtype of IBD, Crohn’s disease can result in mucosal defects throughout the entire GI tract. Severe complications such as fistulas, strictures, and obstruction may also interfere with nutrient absorption. Active inflammation shortens the contact time of nutrients and intestinal mucosal surface, which interferes with absorption of amino acids, and contributes to the exacerbation of malabsorption and the patient’s sarcopenic state [21]. Sarcopenia may be a direct result of malabsorption or an indirect consequence of systemic inflammatory cascade. Chronic inflammation may also have a role in the development of sarcopenia. The systemic elevation of proinflammatory cytokines such as interferon IFN-γ, IL-1, IL-4, and tumor necrosis factor TNF-α are associated with protein catabolism and reduced muscle protein synthesis, potentially through inhibition of the anabolic mTORC1 pathway [22,23]. Consistent with our results, variables associated with active inflammation such as CRP, PLT, and WBC were associated with a higher likelihood of sarcopenia. 

The present study also identified several anthropometric measures, such as BMI and height, as strong predictors for sarcopenia. Interestingly, a taller stature was associated with an increased risk of sarcopenia. The common notion denotes that increased height is commonly associated with increased muscles mass. However, in chronic inflammatory diseases, such as IBD, muscle mass may not necessarily increase with height, instead a negative correlation could be observed. Another possible explanation is of ethic disparity; since this study was conducted in Shanghai, China, only Chinese patients were included in the present cohort. Asians appear to have lower height-adjusted muscle mass values compared to Caucasians [24]. Due to the small sample size of our study, the majority of patients diagnosed with sarcopenia were male (92.4%); therefore, height could also be influenced by the uneven gender distribution. 

Although the etiology of sarcopenia remains elusive, its prognostic value has been confirmed by several studies. Multiple studies have demonstrated that sarcopenia in IBD has been associated with poor outcomes, especially in patients undergoing surgery [25,26,27]. Studies have also shown that the presence of sarcopenia positively correlates with disease severity and prolonged hospital stay or rehospitalization in patients with IBD. Early identification of sarcopenia can sway treatment decisions involving early and active intervention for both disease activity and sarcopenia. Nutritional intervention has been proven effective in maintaining muscle mass and increasing muscle strength and function in sarcopenia, which can be counteracted with adequate dietary protein intake of 1.2–1.5 g/kg/day, especially during active-phase IBD. The role of pharmacological intervention in improving sarcopenia in IBD patients has also received increasing attention. Infliximab improves muscle mass and muscle strength in CD patients after 24 weeks with a significant reduction in IL-6 [28]. Recent publication of Selecting Therapeutic Targets in Inflammatory Bowel Disease (STRIDE)-II has advocated not only the importance of clinical remission and mucosal healing but also the reestablishment of physical function and improvement of QOL or ADL as long-term treatment goals [29]. These statements suggest the important role of sarcopenia in the management of IBD. 

Sarcopenia can be assessed via various methods, including imaging techniques such as computed tomography (CT) and dual-energy X-ray absorptiometry (DXA). Comparatively, CT provides a direct estimate of muscle mass, while DXA provides an indirect estimate of lean mass. However, measurement of muscle mass is insufficient in the determination of muscle function. Grip strength and walking speed can collectively provide a more well-rounded assessment of the disease state [30]. Skeletal muscle signal intensity in MRI T1-weighted images has also been investigated to provide diagnostic reference [31]. To date, there is no consensus as to the gold standard diagnostic modality for sarcopenia [32]. Therefore, the reported incidence in IBD patients ranges from 20–70%, which may be due to differently defined cut-off values across different patient populations, patient ethnicities, and disease statuses, as well as the use of different diagnostic modalities. 

The present study screened and identified an optimal machine learning algorithm, the LightGBM, with a high predictive performance for identifying sarcopenia in Crohn’s disease. The variables used are easily accessible in both inpatient and outpatient settings. Crohn’s disease patients require routine follow-up with periodic disease surveillance. However, abdominal CT is usually ordered once every 6–12 months due to radiation exposure. Apart from the inconvenience of performing an imaging study during every visit, a patient’s sarcopenic state can only be determined based on measurement of skeletal muscle mass performed by a trained radiologist or automated segmentation system, which may not be readily available. The proposed machine learning model can provide an alternative modality to assess sarcopenia in Crohn’s disease patients at shorter intervals. This information could also be additive to evaluating patient response to treatment and disease activity. 

### Study Limitations

Although the present study employed the cross-sectional CT images at L3 of CD patients for accurate measurement of SMM and SMI, there are several limitations. Due to the retrospective nature of the study, muscle strength and physical performance of the patient at the point of study inclusion cannot be assessed. Radiological calculation of skeletal muscle mass remains a rather monotonous assessment of sarcopenia. Secondly, the chronic disease nature of CD makes it difficult to consider disease duration and previous treatment received as confounding factors when determining a patient’s sarcopenic state.

## 5. Conclusions

Accumulating evidence has demonstrated the diagnostic and prognostic significance of sarcopenia in IBD patients [33]. Early identification and appropriate intervention of sarcopenia may accelerate clinical remission and mucosal healing. The present study proposes a noninvasive predive model based on anthropometric data and clinical variables to predict the presence of sarcopenia in CD patients. The LightGBM model may facilitate a population management strategy with early identification of sarcopenia, while also providing guidance for nutritional support and an alternative surveillance modality for long-term patient follow-up.

## Figures and Tables

**Figure 1 ijerph-20-00656-f001:**
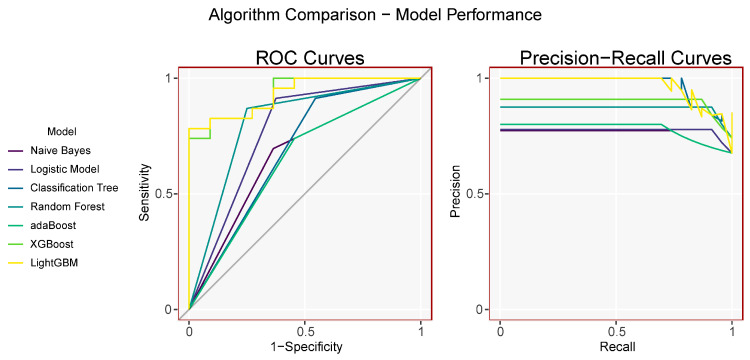
Receiver operating characteristic curves and precision-recall curves of the seven different machine learning models.

**Figure 2 ijerph-20-00656-f002:**
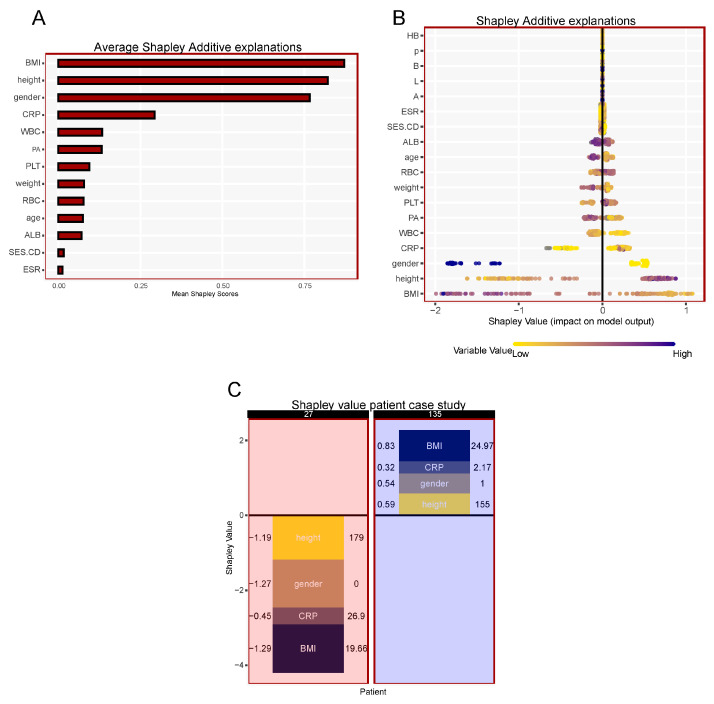
Ranking of the importance of the top 14 variables (**A**) with Shapley additive explanations (SHAP) summary plot (**B**) in the LightGBM model for predicting sarcopenia in Crohn’s disease. Shapley values of the top four variables based on the LightGBM model predicts sarcopenia in representative patient samples (**C**). BMI: body mass index; CRP: C-reactive protein; WBC: white blood count; PA: prealbumin; PLT: platelet; RBC: red blood count; ALB: albumin; SES-CD: simple endoscopic score for Crohn’s disease; ESR: erythrocyte sedimentation rate; HB: hemoglobin; p: perianal disease; B: disease behavior; L: disease location; A: age at diagnosis.

**Table 1 ijerph-20-00656-t001:** Baseline clinical, anthropometric, and radiological characteristics of the included patients.

	Overall Population *n* = 167	Sarcopenia *n* = 118	No Sarcopenia *n* = 49	*p*-Value
Demographics				
Gender				<0.001
Female	40 (24.0%)	9 (7.6%)	31 (63.3%)	
Male	127 (76.0%)	109 (92.4%)	18 (36.7%)	
Age	36.1 (14.3)	32.6 (12.4)	44.6 (15.3)	<0.001
Montreal Classification				
Age at diagnosis				<0.001
A1 (<17 years old)	6 (3.6%)	6 (5.1%)	0 (0.00%)	
A2 (17–40 years old)	105 (62.9%)	85 (72.0%)	20 (40.8%)	
A3 (>40 years old)	56 (33.5%)	27 (22.9%)	29 (59.2%)	
Location				0.186
L1 (Ileal)	76 (45.8%)	48 (40.7%)	28 (58.3%)	
L2 (Colonic)	14 (8.4%)	10 (8.5%)	4 (8.3%)	
L3 (Ileocolonic)	71 (42.8%)	56 (47.5%)	15 (31.2%)	
L4 (Upper Disease)	5 (3.0%)	4 (3.4%)	1 (2.1%)	
Disease Behavior				0.446
B1 (nonstricturing, nonpenetrating)	57 (34.3%)	44 (37.3%)	13 (27.1%)	
B2 (stricturing)	86 (51.8%)	58 (49.2%)	28 (58.3%)	
B3 (penetrating)	23 (13.9%)	16 (13.6%)	7 (14.6%)	
Perianal Disease				0.11
Yes	124 (74.3%)	35 (29.7%)	8 (16.3%)	
No	43 (25.7%)	83 (70.3%)	41 (83.7%)	
Endoscopic Scores				
SES-CD	6.47 (5.9)	6.81 (6.4)	5.60 (4.5)	0.191
Laboratory Parameters				
White blood cell, WBC (×10^9^)	6.41 (2.5)	6.70 (2.7)	5.72 (1.8)	0.007
Red blood cell, RBC (×10^12^)	4.44 (0.7)	4.54 (0.6)	4.22 (0.7)	0.005
Hemoglobin, Hb (g/L)	123 (27.7)	123 (20.9)	124 (39.8)	0.823
Platelet, PLT (×10^9^)	274 (94.6)	284 (91.9)	250 (97.7)	0.039
Albumin, ALB (g/L)	37.8 (5.8)	37.9 (6.1)	37.7 (5.2)	0.855
Prealbumin, PA (mg/L)	183 (60.4)	179 (57.8)	193 (66.3)	0.211
Erythrocyte sedimentation rate, ESR (mm/h)	22.3 (24.4)	21.9 (23.5)	23.3 (26.8)	0.772
C-reactive protein, CRP (mg/L)	19.0 (25.0)	20.8 (26.1)	14.4 (21.4)	0.122
Anthropometrics				
Height, H (cm)	169 (8.3)	172.3 (7.1)	162.65 (7.1)	<0.001
Weight, W (kg)	57.3 (9.9)	57.02 (9.3)	57.9 (11.3)	0.643
Body mass index, BMI (kg/m^2^)	19.9 (2.9)	19.13 (2.4)	21.7 (3.2)	<0.001
Body Composition				
Skeletal Muscle Mass, SMM (cm^2^)	115.7 (25.7)	117.4 (22.1)	111.4 (32.7)	0.171
Skeletal Muscle Index, SMI (cm^2^/m^2^)	40.0 (7.6)	39.4 (6.4)	41.6 (9.8)	0.147
Visceral Adipose Tissue, VAT (cm^2^)	64.0 (50.8)	53.3 (45.0)	89.7 (55.2)	<0.001
Visceral Adipose Index, VAI (cm^2^/m^2^)	22.4 (17.9)	17.7 (14.7)	33.7 (19.9)	<0.001
Subcutaneous Adipose Tissue, SAT (cm^2^)	79.2 (50.2)	64.5 (42.8)	114.7 (49.4)	<0.001
Subcutaneous Adipose Index, SAI (cm^2^/m^2^)	28.0 (18.4)	21.6 (13.8)	43.44 (19.0)	<0.001

**Table 2 ijerph-20-00656-t002:** Performance of the prediction models generated by seven machine learning algorithms. F1: F1 score; MCC: Matthews correlation coefficient; AUC: area under curve; AUPRC: area under the precision-recall curve; TP: true positive; FP: false positive; FN: false negative; TN: true negative; PPV: positive predictive value; NPV: negative predictive value.

Diagnostic Model	Accuracy	Sensitivity	Specificity	Precision	F1	MCC	AUC	AUPRC	TP	FP	FN	TN	PPV	NPV
Naive Bayes	0.6765	0.6364	0.6957	0.5000	0.5600	0.3156	0.6660	0.7779	7	4	7	16	0.6364	0.6957
Logistic Model	0.8387	0.6250	0.9130	0.7143	0.6667	0.5631	0.7690	0.8687	5	3	2	21	0.6250	0.9130
Classification Tree	0.7647	0.4545	0.9130	0.7143	0.5556	0.4253	0.6838	0.7730	5	6	2	21	0.4545	0.9130
Random Forest	0.8387	0.7500	0.8696	0.6667	0.7059	0.5973	0.8098	0.8969	6	2	3	20	0.7500	0.8696
adaBoost	0.6765	0.5455	0.7391	0.5000	0.5217	0.2786	0.6423	0.7584	6	5	6	17	0.5455	0.7391
XGBoost	0.7941	0.6364	0.8696	0.7000	0.6667	0.5195	0.9130	0.9650	7	4	3	20	0.6364	0.8696
LightGBM	0.8235	0.7273	0.8696	0.7273	0.7273	0.5968	0.9328	0.9701	8	3	3	20	0.7273	0.8696

## Data Availability

Original data can be made available via formal acquisition to the corresponding author.

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
