# Peer review of "Machine Learning Model in Predicting Sarcopenia in Crohn’s Disease Based on Simple Clinical and Anthropometric Measures"

_ijerph, 2022, doi:10.3390/ijerph20010656_

Round 1

Reviewer 1 Report

The original article from Tseng et al. aimed to validate a model to predict sarcopenia in Crohn’s disease.

General comments

Although the subject is of interest, the article needs to be rewritten to be more focused on the objective. The conclusions are quite disappointing since BMI and CRP are two of the main determinants of sarcopenia, which is already known. What is the added value of the model in this context? It should be discussed. The model is not externally validated.

Specific comments

Introduction

The introduction section focused on sarcopenia. A paragraph on the interest of machine learning models should be added. The link between lines 65-67 and the aim of the study should be explained more clearly.

Material & Methods

Lines 86-91, the choice of variables should be justified

Lines 97-98, who performed the segmentation? Is there a double check?

Lines 106-108, the citation is inadequate, the selected cut-off should be validated

Lines 110-132, references for each model are missing

Results

Text in paragraph 3.1 is redundant with the table 1. Please keep only the Table 1

Figure 1 is not necessary

Figure 4, you need to explain abbreviations

The quality of Figure 2 is insufficient

What is the performance of BMI alone?

There are too many figures. There are redundancies between figures

Discussion

Discussion should be rewritten since this section is not focused on the objective of the study. Some parts are only generalities, e.g. lines 271-284, lines 288-304, lines 305-317

Lines 313-314: it is discordant with lines 64-65

Height as a factor determinant for sarcopenia in this study, which is surprising. This result should be discussed.

Reviewer 2 Report

In this paper, the authors present a study, using their own words, ‘[…] aimed at investigating the different diagnostic performance of different machine learning models in identifying sarcopenia in Crohn’s Disease’.

After a detailed review, I have some questions and considerations. For that reason, I propose to a revision.

Please, see my comments in the attached pdf file.

Round 2

Reviewer 1 Report

1) Is the Light GMB model made available for scientific community?

2) In Figures 2 & 3, it should be mentioned that it is applicable to the Light GMB model

3) The Figures 2, 3, 4 could be associated in a single Figure with Subfigures. The text related to this figure can be more concise

4) The discussion can be more concise (paragraph lines 301-319, line 332-348, lines 349-361 are not a discussion of the results but rather generalities on the topi). Shorten these partes and discussed your results in relation with the litterature

5) There are no references to justify the links between height and sarcopenia

Reviewer 2 Report

Thank you for the responses. I just have some minor comments:

I would choose the 200 words abstract. Anyway, I differ this decision to the authors.

I was not able to find the paragraph at the end of the introduction section explaining how the rest of the paper is organized. Please, revise it. 
